# BNT162b2 mRNA vaccine elicited antibody response in blood and milk of breastfeeding women

Michal Rosenberg-Friedman[1,2,7], Aya Kigel[3,7], Yael Bahar[3], Michal Werbner[4], Joel Alter[5], Yariv Yogev[1,2], Yael Dror[3], Ronit Lubetzky[2,6], Moshe Dessau [5], Meital Gal-Tanamy[4], Ariel Many[1,2] & Yariv Wine [3✉]

The importance of breastmilk in postnatal life lies in the strong association between breastfeeding and the reduction in the risk of infection and infection-related infant mortality. However, data regarding the induction and dynamics of breastmilk antibodies following administration of the Pfizer-BioNTech BNT162b2 COVID-19 mRNA vaccine is scarce, as pregnant and lactating women were not included in the initial vaccine clinical trials. Here, we investigate the dynamics of the vaccine-specific antibody response in breastmilk and serum in a prospective cohort of ten lactating women who received two doses of the mRNA vaccine. We show that the antibody response is rapid and highly synchronized between breastmilk and serum, reaching stabilization 14 days after the second dose. The response in breastmilk includes both IgG and IgA with neutralization capacity.

[1] Department of Obstetrics and Gynecology, Lis Maternity & Women's Hospital, Tel Aviv Sourasky Medical Center, Tel Aviv, Israel. [2] Sackler Faculty of Medicine, Tel Aviv University, Tel Aviv, Israel. [3] The Shmunis School of Biomedicine and Cancer Research, The George S. Wise Faculty of Life Sciences, Tel Aviv University, Tel Aviv, Israel. [4] Molecular Virology Lab, The Azrieli Faculty of Medicine, Bar-Ilan University, Safed, Israel. [5] The Laboratory of Structural Biology of Infectious Diseases, The Azrieli Faculty of Medicine, Bar-Ilan University, Safed, Israel. [6] Department of Pediatrics, Dana Dwek Children's Hospital, Tel Aviv Sourasky Medical Center, Tel Aviv, Israel. [7] These authors contributed equally: Michal Rosenberg-Friedman, Aya Kigel.
✉email: yarivwine@tauex.tau.ac.il

An accelerated vaccination campaign with the Pfizer-BioNTech BNT162b2 COVID-19 mRNA vaccine (hereinafter mRNA vaccine) was initiated in Israel in December 2020, and as of April 2021 nearly 50% of the adult population had received two doses of the vaccine[1]. The vaccine campaign initially targeted high-risk populations (≥60 years old and healthcare providers)[2], but it was soon expanded to include other sectors of the population, including lactating women[3]. Notwithstanding the reported high efficacy of this vaccine[4] and evidence for the generation of viral-specific antibodies in the breastmilk of women with COVID-19[5–7], there is little data available on the efficacy of the vaccine in lactating women or its potential benefits in neonatal protection via the passive transfer of vaccine-specific antibodies in breastmilk[8]. To date, two reports have described the breastmilk antibody response following an mRNA-vaccination[9,10] (one with the Pfizer vaccine and the other with the Pfizer or the Moderna mRNA-1273 vaccine), but there is still vital information lacking on the neutralization capacity of breastmilk antibodies, on the dynamics of the antibody response in breastmilk in comparison to the response in blood, and on whether antibody levels are similar to those elicited in lactating women following the administration of other vaccines. This knowledge gap is preventing global health authorities from making concrete recommendations regarding vaccination during lactation.

We thus investigated the temporal dynamics of the breastmilk and serum antibody response in a prospective cohort of ten lactating healthcare providers, mean age 34.6 (range 30−38), who received the first dose of the mRNA vaccine approximately five months postpartum (mean 154 days, range 68−382) and the second dose 21 days later (Supplementary Table 1). Importantly, we also determined the neutralization capacity of the breastmilk antibodies and compared their levels to those that were elicited by maternal vaccination with the tetanus-diphtheria-acellular pertussis (TDaP) vaccine.

Here, we report that the antibody response following the mRNA vaccine is rapid and highly synchronized between serum and breastmilk and that antibodies in breastmilk exhibit neutralization capacity. Moreover, the breastmilk antibody levels elicited by two mRNA vaccine doses were found to be similar to those induced by the TDaP vaccine. These findings suggest that vaccination during breastfeeding could confer protection during early infancy.

## Results

**IgG and IgA in serum and breastmilk following BNT162b2 mRNA vaccination.** To obtain calculated endpoint titers for IgG and IgA in serum and breastmilk dyads against the SARS-CoV-2 spike and receptor-binding domain (RBD) proteins, serial dilution ELISAs were run on days 7 and 14 after the first (designated 1D7 and 1D14, respectively) and second (designated 2D7 and 2D14, respectively) vaccine doses (Supplementary Figs. 1 and 2). We found that the spike (Fig. 1) and RBD-specific (Supplementary Fig. 3) antibody responses in serum and breastmilk were rapid and synchronized for IgG and IgA.

On 1D7, the spike-specific endpoint titers of IgG and IgA in the serum and breastmilk had not increased significantly above the titers in the control group (pre-pandemic serum and breastmilk), and the first significant increase in antibody titers was evident on 1D14. The titers for the vaccinated group remained significantly higher than those of the control group, with spike-specific endpoint titers peaking on 2D7, followed by a non-significant decrease on 2D14. A similar trend was observed for the RBD-specific IgG titers (Supplementary Fig. 3). The RBD-specific IgA titers, in both serum and breastmilk, reached significant levels vs. control titers on 2D7, suggesting that the accumulation of IgA antibodies against the restricted region of the spike (i.e., RBD) was slightly delayed compared to IgG, both in serum and breastmilk.

Since all participants had received the TDaP vaccine during the third trimester[11], we were able to compare the breastmilk endpoint titers for IgG and IgA elicited by the mRNA vaccine on 2D14 to tetanus toxoid (TT)-specific antibody titers in the same participants. We found that the spike- and RBD-specific IgG and IgA titers did not differ significantly from the TT-specific antibody titers.

**Vaccine-specific IgG:IgA molar ratio in breastmilk and serum.** We then evaluated the vaccine-specific IgG:IgA ratio in breastmilk and serum by interpolating the antibody concentrations from IgG and IgA standard curves. The serum antibody response was dominated by IgG, and the IgG:IgA molar ratio was significantly higher in serum than in breastmilk at all four time points (Supplementary Fig. 4a) and this ratio did not exhibit significant temporal change (Supplementary Fig. 4b). The IgG:IgA ratio in breastmilk indicated that the vaccine-specific response was dominated by IgA at all time points; however, the IgG:IgA ratio significantly increase at the 2D7 and 2D14 timepoints suggesting a temporal increase in IgG levels (Supplementary Fig. 4c), as previously described following respiratory syncytial virus immunization[12].

**Temporal dynamics of vaccine-specific antibodies.** To better understand the temporal dynamics of the antibody response following mRNA vaccination, we calculated the fold-change of anti-spike (Fig. 2), anti-RBD (Supplementary Fig. 5) endpoint titers of IgG and IgA at each time point over the endpoint titers on the 1D7 time point. The reference point for determining the changes in the accumulated production of antibodies in serum and breastmilk was taken as 1D7. The vaccine-specific IgG and IgA titers in serum and breastmilk increased substantially at each time point, but the increase in fold-change halted 14 days following the second dose, indicating that production was stabilized on 2D14 (the fold-change on 2D14 was not significantly higher than that at the preceding time point). Noteworthy and limited to spike- and RBD-specific IgA in breastmilk, the fold-change above the endpoint titer on 1D7 and on 2D14 declined significantly, suggesting that the production rate of IgA in breastmilk was in a decline 14 days following the second dose. Calculation of the fold-change compared to the preceding time point provided information on the dynamics of the antibody response following each dose for the anti-spike and RBD antibodies (Supplementary Fig. 6). The antibody levels peaked seven days following each vaccine dose, showing the primary and boost the effect of each dose on the antibody titer.

**Neutralization of breastmilk and purified breastmilk IgG/IgA.** The SARS-CoV-2 neutralization capacity of breastmilk was determined using spike-bearing pseudovirus neutralization assay. First, the cell toxicity of breastmilk was assessed showing that filtered breastmilk does not have cell toxicity attributes (Supplementary Fig. 7) thus, all subsequent neutralization assays were conducted using filtered breastmilk samples.

Next, breastmilk $ID_{50}$ was determined for all 10 participants and 5 control samples (Fig. 3a). We found that all breastmilk samples at 2D7 exhibit inhibition activity (mean $ID_{50} = 17$, range 3.2−52.2, breastmilk reciprocal dilution) and overall, the $ID_{50}$ values were significantly higher compared to the control group (Fig. 3b). Of note, a recent publication[6] reported that only 62% of

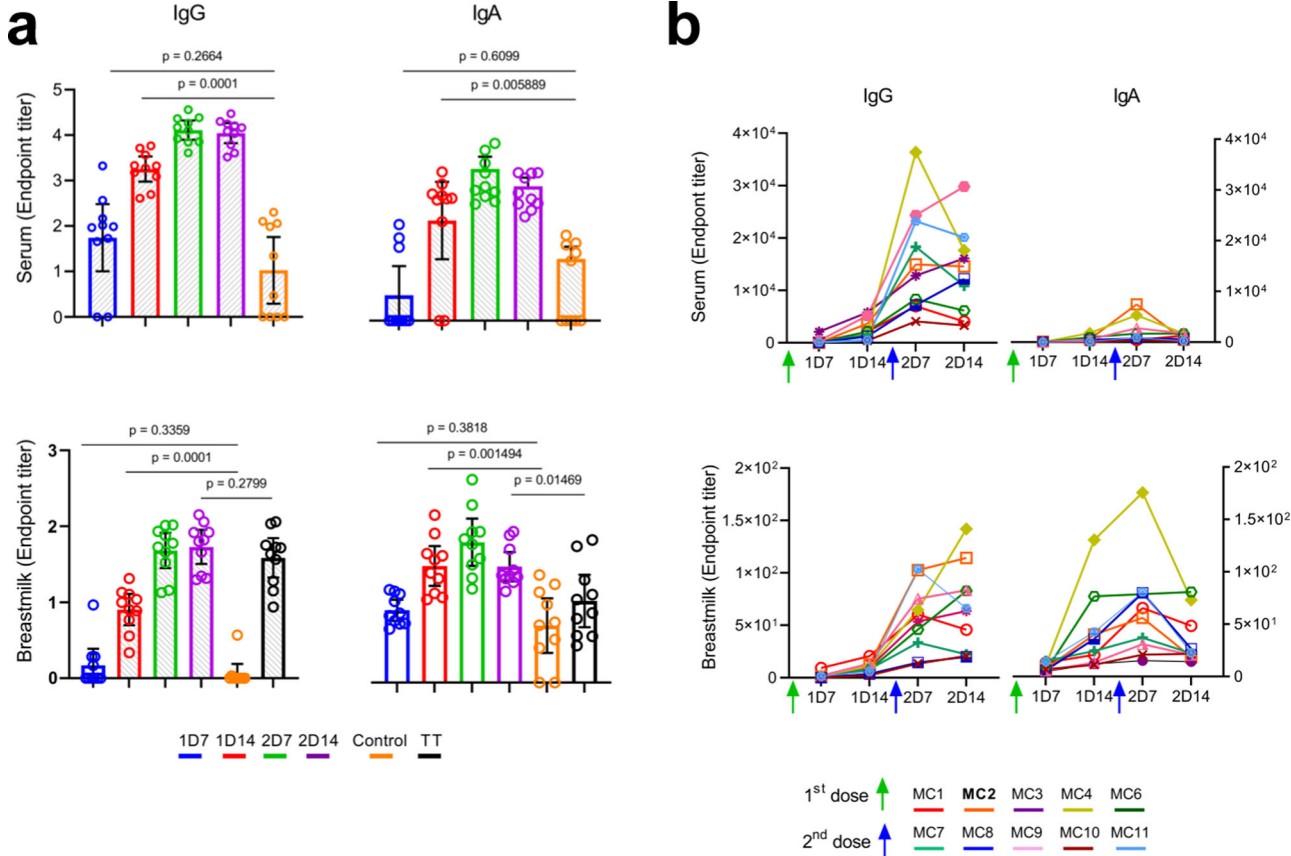

**Fig. 1 Endpoint titers of spike-specific IgG and IgA. a** Endpoint titers were interpolated by applying a four-parameter logistic curve on reciprocal dilution series for IgG and IgA for all serum and breastmilk samples at four time points (by color) and for tetanus toxoid (TT)-specific IgG and IgA in breastmilk. *P* values were determined with an unpaired, two-sided Mann−Whitney U-test after applying Bonferroni correction; $P < 0.0125$ was considered statistically significant. Results are presented as geometric means and 95% confidence intervals. *Y*-axis units are endpoint titers on a logarithmic scale. Control serum ($n = 10$) and breastmilk samples ($n = 10$) were obtained from healthy women before the COVID-19 pandemic. **b** Endpoint titers per participant, each shown in a different color. Green and blue arrows indicate the time points of administration of the first ($t = 0$) and second ($t = 21$) vaccine doses, respectively. *Y*-axis units are endpoint titers on a linear scale. Source data are provided as a Source Data file.

breastmilk samples (containing antibodies) from women with COVID-19 were found to have neutralization capacity in vitro.

Next, we measured the neutralization of IgG and IgA antibodies in breastmilk. Breastmilk IgG and IgA were purified using protein G and peptide M, respectively and were tested independently for neutralization capacity. For purified IgG at the concentration of 50 μg/ml (final concentration 33 nM), 4 out of 10 breastmilk 2D7 samples exceeded the 50% inhibition while none of the IgA samples, at 50 μg/ml (final concentration 13 nM), exceeded the 50% inhibition activity (Fig. 3c). Nevertheless, the normalized inhibition activity (per 5 μg antibody) of IgG and IgA purified from breastmilk 2D7 samples was on average 21-fold and 4-fold higher than the inhibition activity of IgG and IgA (respectively) isolated from control samples (Fig. 3d).

## Discussion
The importance of breastfeeding in early infancy is highlighted by the strong correlation between breastfeeding and the overwhelming decline in risks of infection and infection-associated morbidity and mortality[13,14]. In particular, breastfeeding has been associated with a decrease in the number of cases of respiratory illness[15] and a decreased risk of hospitalization for respiratory diseases[16,17]. The BNT162b2 COVID-19 vaccine rollout in Israel provided a unique opportunity to study the dynamics of breastmilk antibodies during

the primary and secondary immune response. The importance of the findings extends far beyond the current vaccine campaign, since they provide new insights regarding the maternal immune response following vaccination, specifically vaccination during lactation. Our data indicate that the vaccine-specific antibody response following administration of the mRNA vaccine is highly synchronized between serum and breastmilk, with a wave of antibody production that peaks 7 days following each vaccine dose. Furthermore, an analysis of the dynamics shows that breastmilk levels of IgG peaked 14 days following the second vaccine dose; breastmilk levels of IgA followed a similar pattern, but showed a slight decrease at that time point. The antibody levels induced in breastfeeding women vaccinated with the mRNA vaccine were similar to those induced by TDaP vaccine that was given during pregnancy, suggesting that the mRNA vaccine-specific antibodies may exhibit similar protective potential as observed for TDaP specific antibodies. Importantly, the neutralization capacity of breastmilk was evident in all women who participated in the study. Of note, the percent inhibition observed for breastmilk IgA was lower than that for IgG however the breastmilk IgA concentration was an order of magnitude higher than IgG suggesting that the neutralization activity may rely predominantly on IgA. Moreover, although IgG and IgA were used in equal concentrations (50 μg/ml) in the pseudovirus neutralization assays, IgA molarity is approximately 2.5-fold lower than that of IgG, as IgA exists as a secretory dimer in breastmilk (~380 kDa).

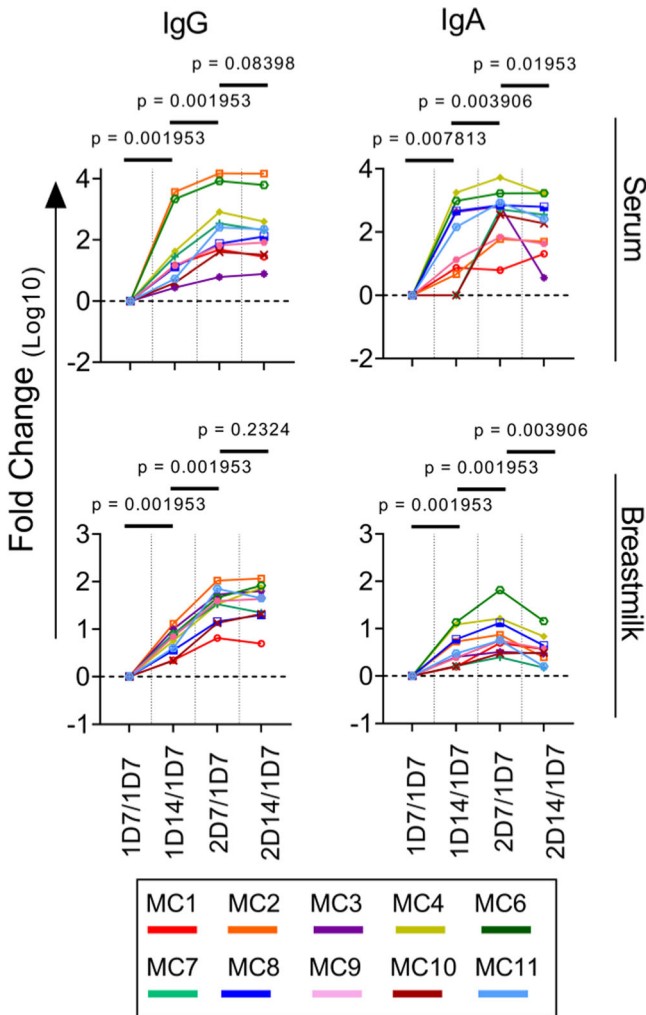

**Fig. 2 Dynamics of the spike-specific antibody response.** Fold-changes in antibody titers compared to the first time point are plotted by patient. Comparisons between the fold-change in spike-specific IgG and IgA antibodies were made using the two-tailed Wilcoxon matched-paired signed-rank test; $P < 0.0125$ was considered statistically significant following Bonferroni correction (accounting for multiple comparisons). Y-axis units are fold-change on a logarithmic scale. Source data are provided as a Source Data file.

The molar differences can explain the reduced neutralization capacity of IgA compared to that of IgG. Still, applying neutralization assay at a single concentration provides interesting insights on how the neutralization activity of breastmilk IgG and IgA is varied between participants. For example, MC3 and MC8 show very low neutralization activity for IgA while exhibiting high neutralization activity for IgG. Noteworthy, breastmilk (as opposed to serum) is a renewable resource of antibodies as it is consumed daily by the lactating infant. Thus, while it may seem that neutralization activity is relatively low, the continuous consumption of breastmilk can provide an additive protection effect.

In summary, the study provides evidence for the rapid production of vaccine-specific antibodies, both IgA and IgG. Moreover, neutralization capacity for breastmilk was observed in all samples. This study also indicates the potential protection conferred on breastfed infants by the administration of the mRNA vaccine to the breastfeeding mother. Nonetheless, temporal studies for longer periods with a larger cohort size are needed to further elucidate the persistence of the breastmilk antibodies. In addition, real-world evidence regarding the actual protection that

breastmilk vaccine-specific antibodies confer on the neonate should be further evaluated by surveillance of COVID-19 incidents in infants that are breastfed by vaccinated women.

**Limitation of study.** While we report on the $ID_{50}$ values for all breastmilk samples obtained at the 2D7 time point, the neutralization assay for purified breastmilk IgG and IgA could be carried out only at a single concentration (50 μg/ul) thus, the $IC_{50}$ could not be calculated. This is due to restricted volumes of breastmilk donated by study participants. Thus, the insights regarding the neutralization capacity of either breastmilk IgA or IgG are limited. Specifically, as IgA at the concentration of 50 μg/ml did not exceed the 50% inhibition, we are careful with any statement regarding the neutralization capacity of IgA. Still, the data demonstrates the participants variability in the context of the neutralizing activity of IgG and IgA.

## Methods

**Human subjects, sample collection, and processing.** The cohort comprised 10 lactating healthcare providers who received two doses of the Pfizer-BioNTech BNT162b2 COVID-19 mRNA vaccine (mRNA vaccine) with a 21-day interval between the first and second doses. Control serum and breastmilk samples were obtained from 10 healthy women collected prior the COVID-19 pandemic. All participants provided informed consent for the use of their data and clinical samples for the purposes of the present study. Sample collection was performed under institutional review board approvals number 0002269-4 and 0002757-1 given at Tel Aviv University and under ethical approval number 1088-20-TLV given at Tel Aviv Sourasky Medical Center. Breastmilk and blood dyads were collected from the COVID-19 vaccinees into BD vacutainer K2 EDTA collection tubes and sterile containers, respectively. Sample dyads were collected at four time points, namely, 7 and 14 days following the first and second vaccine doses (designated 1D7, 1D14, 2D7, and 2D14, respectively).

Isolation of plasma from whole blood was performed by density gradient centrifugation, using Uni-SepMAXI$^+$ lymphocyte separation tubes (Novamed) according to the manufacturer's protocol. The breastmilk aqueous phase was separated from whole milk by centrifugation in 50 ml conical tubes at $500 \times g$, swinging bucket, room temperature (RT), 20 min, acceleration = 9, brake = 1. For each sample, the upper lipid layer was discarded, and the aqueous phase was transferred to a clean 50 ml tube. All serum and breastmilk aqueous phase samples were stored at −20 °C.

**Expression and purification of recombinant protein.** The plasmids for expression of recombinant SARS-CoV-2 receptor-binding domain (RBD) and spike protein were kindly provided by Dr. Florian Krammer, Department of Microbiology, Icahn School of Medicine at Mount Sinai, New York, NY, USA. The RBD sequence is based on the genomic sequence of the first virus isolate, Wuhan-Hu-1, which was released on 10 January 2020[18]. The plasmids for the expression of recombinant human angiotensin I converting enzyme 2 (hACE2) were kindly provided by Dr. Ronit Rosenfeld, Israel Institute for Biological Research (IIBR). The cloned region encodes amino acids 1-740 of hACE2 followed by 8xHis-tag and a Strep Tag at the 3′ end, cloned in a pCDNA3.1 backbone. Recombinant RBD and hACE2 were produced in Expi293F cells (ThermoFisher Scientific) by transfection of the cells with a purified mammalian expression vector using an ExpiFectamine 293 Transfection Kit (Thermo Fisher Scientific), according to the manufacturer's protocol, and as described previously[18]. Supernatants from transfected cells were purified on a HisTrap affinity column (GE Healthcare) using a two-step elution protocol with 5 column volumes (CV) of elution buffer supplemented with 50 mM imidazole in phosphate-buffered saline (PBS), pH 7.4, followed by 250 mM imidazole in PBS, pH 7.4, for RBD and spike proteins and by 500 mM imidazole in PBS for hACE2. Elution fractions containing clean recombinant proteins were pooled and dialyzed using Amicon Ultra (Mercury) cutoff 10 K against PBS (pH 7.4). Dialysis products were analyzed by 12% SDS–PAGE for purity, and concentration was measured using Take-5 (BioTek Instruments).

**Endpoint titers of spike and RBD-specific IgA, IgG in serum and breastmilk.** Serum and breastmilk IgG and IgA antibody endpoint titers were determined by enzyme-linked immunosorbent assay (ELISA). Spike$^+$ and RBD$^+$ Ig in serum and breastmilk were determined using half-area 96-well ELISA plates (Greiner Bio-One) that had been coated overnight at 4 °C with 2 μg/ml RBD or spike proteins in PBS (pH 7.4). Thereafter, the coating solution was discarded, and the ELISA plates were blocked with 150 μl of 3% w/v skim milk in PBS for 1 h at 37 °C. After discarding the blocking solution, duplicates of 36 μl of serum diluted 1:100 or 36 μl of breastmilk diluted 1:1 in 3% w/v skim milk in PBS were added to the first row of the coated plate. Dilutions were carried out with a threefold dilution factor, and the plates were incubated for 1 h at RT. Then, plates were washed three times with 0.05% PBS-Tween 20 (PBST) and incubated for 1 h at RT with horseradish peroxidase (HRP) conjugated anti-human IgG)Jackson Immunoresearch, #CAT

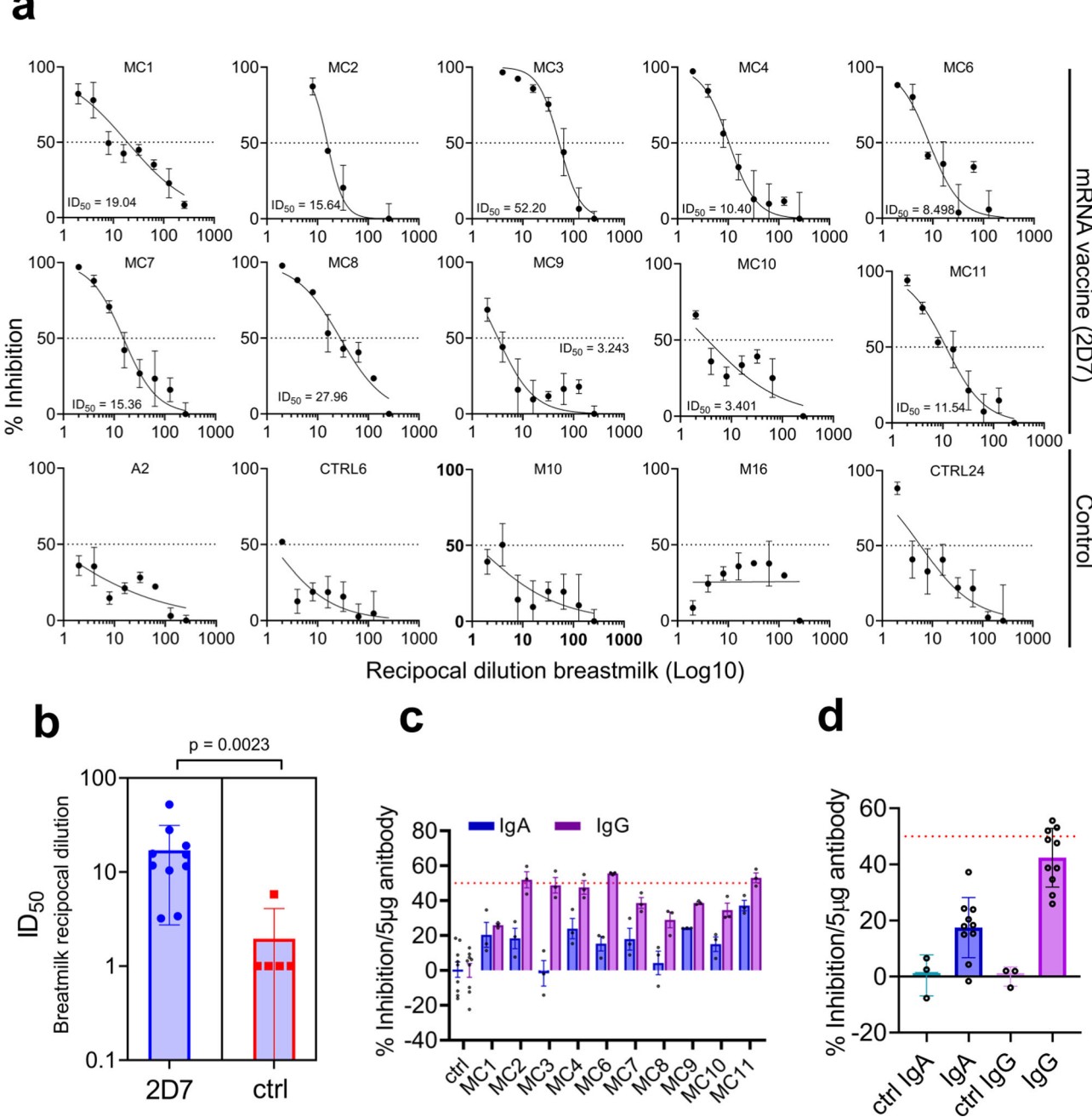

**Fig. 3 Neutralization of breastmilk and purified breastmilk IgG and IgA. a** SARS-CoV-2 spike bearing pseudovirus neutralization assay on serial dilutions of filtered mRNA-vaccine breastmilk samples collected at 2D7 ($n = 10$) and of pre-pandemic breastmilk control samples ($n = 5$). Breastmilk ID$_{50}$ is indicated in each plot as calculated by the nonlinear regression [inhibitor vs. response] model. Each measurement was carried out in triplicates. Mean and s.d. are indicated for each dilution. Mean and error bar for s.d. are indicated. **b** Comparison of ID$_{50}$ values of breastmilk from 2D7 ($n = 10$) and control samples ($n = 5$). P values were determined with an unpaired, two-sided Mann−Whitney U-test; $P < 0.05$ was considered statistically significant. Mean and 95% CI are indicated. **c** Purified IgG and IgA from breastmilk collected at 2D7 ($n = 10$) and pre-pandemic control samples ($n = 3$) (concentration of 50 µg/ml, 33 nM, 13 nM for IgG and IgA, respectively) were subjected to a spike-bearing pseudovirus neutralization assay; percent inhibition was calculated by measuring GFP-positive HEK-293 cells stably expressing hACE2. The number of GFP-positive cells was normalized and converted to a neutralization percentage. Each sample was measured in triplicate, and bar graph plots present mean values ± SEM. Precent inhibition was normalized per 5µg antibody. The horizontal dashed red line indicates 50% inhibition. **d** Bar plot summarizing the percent inhibition 2D7 breastmilk IgG and IgA ($n = 10$) in comparison with IgG and IgA from pre-pandemic control samples. The horizontal dashed red line indicates 50% inhibition. Mean and 95% CI are indicated. Source data are provided as a Source Data file.

109035003)/anti-human IgA (Jackson Immunoresearch, #CAT 109035011) (25 µl, 1:5000 ratio in 3% w/v skim milk in PBS). Plates were then subjected to three washing cycles with 0.05% PBST, and developing was carried out by adding 25 µl of 3,3′,5,5′-tetramethylbenzidine (TMB), followed by quenching with 25 µl of 1 M sulfuric acid. Plates were read using the Epoch Microplate Spectrophotometer

ELISA plate reader at a wavelength of 450 nm. Endpoint titers were determined as the maximum serum/breastmilk dilution with an O.D.$_{450}$ signal that is 3 standard deviations above background.

The Mann−Whitney test was used to compare continuous variables of two independent groups, and significance was set at $P = 0.0125$ following Bonferroni

correction. The Wilcoxon signed-rank test was used to compare matched samples. All reported *P* values were two-tailed. All statistics were performed with GraphPad Prism 9.0.2 (GraphPad Software).

**Breastmilk IgG and IgA purification**. A volume of 5 ml of the 2D7 breastmilk samples from each participant was diluted threefold in PBS and filtered through a Corning bottle-top vacuum filter system (Merck). IgG was purified by affinity chromatography using 0.5 ml of protein G agarose beads (Millipore) packed in a 5 ml plastic chromatography column (Thermo Scientific). Diluted samples were applied to the protein G affinity column in gravity mode. The column was washed with 10 CVs of PBS, and IgG was eluted using 5 CVs of 100 mM glycine, pH 2.7, and immediately neutralized with 1 M Tris-HCl, pH 8.5. The concentrations of IgG in the elution fractions were measured by Take3™ Micro-Volume Plate, used in BioTek microplate reader (BioTek), and evaluated for IgG purity using 12 % SDS-PAGE. The protein G affinity column flow-through fraction was applied to a peptide M affinity column, where peptide M is a 50 amino acid peptide derived from a streptococcal M protein that binds monomeric and dimeric human IgA of both subclasses (IgA1 and IgA2) with high specificity and affinity[19]. For IgA purification, 1 ml of NHS dry agarose beads (Thermo Scientific) were coupled with peptide M and packed into a 5 ml plastic chromatography column. The flow-through fraction from the protein G affinity column was applied to a peptide M affinity column in gravity mode. The peptide M affinity column was washed with 10 CVs of PBS, and IgA was eluted using 5 CVs of 100 mM glycine, pH 2.7, and immediately neutralized with 1 M Tris-HCl, pH 8.5. For IgG and IgA, elution fractions were merged and dialyzed using VIVASPIN 500 cutoff 50 K (Sartorius Stedim biotech) against PBS. All sample concentrations were measured by Take3™ Micro-Volume Plate, used in your BioTek microplate reader (BioTek), and their purity was evaluated by 12% SDS–PAGE.

**Spike-specific IgG:IgA molar ratio**. The spike-specific IgG:IgA molar ratio in breastmilk and serum at four time points was evaluated using ELISA. Half-well ELISA plates were coated overnight at 4 °C with 2 μg/ml SARS-CoV-2 spike protein in PBS. Thereafter, the coating solution was discarded, and the ELISA plates were blocked with 150 μl of 3% w/v skim milk in PBS for 1 h at 37 °C. After discarding the blocking solution, duplicates of 25 μl of serum diluted 1:100 or 50 μl of breastmilk diluted 1:6 in 3% w/v skim milk in PBS from each time point were added and the plates were incubated for 1 h at RT. To compare IgG and IgA, each plate included a standard, where the micro wells were coated with 2 μg/ml of either protein G or protein M in PBS. Fifty microliters of pure IgG or IgA diluted to 60 μg/ml in 3% w/v skim milk in PBS were added to the first row of the coated plate. Dilutions were carried out with a twofold dilution factor. Then, plates were washed three times with 0.05% PBST and incubated for 1 h at RT with HRP conjugated anti-human IgG /anti-human IgA (25 μl, 1:5000 ratios in 3% w/v skim milk in PBS). Plates were then subjected to three washing cycles with 0.05% PBST, and developing was carried out by adding 25 μl of TMB, followed by quenching with 25 μl of 1 M sulfuric acid. Plates were read using the Epoch Microplate Spectrophotometer ELISA plate reader at a wavelength of 450 nm. Calculation of IgG and IgA molarity was carried out by interpolating sample values from the four-parameter logistic (4PL) standard curve. The interpolated values were then multiplied by the dilution factor (serum 1:100 and breastmilk 1:6), and molar concentrations were used to calculate the IgG:IgA spike-specific molar ratio.

**XTT assay**. XTT (Biological Industries Ltd.) cell-proliferation assay was performed according to the instructions provided by the suppliers. Briefly, HEK293 ACE2-stable cells were plated in 96-well tissue culture plates at a density of 50,000 cells per well in 100 μL of supplemented DMEM and incubated overnight. The following day cells were subjected to various treatments for 3 h followed by medium replacement and incubation for an additional 24 h. Next, the medium was removed, and the cells were incubated with medium containing XTT and the electron-coupling reagent for 2 h. Absorbance was determined at 450 and 630 nm, serving as reference, using the Infinite M1000 PRO (TECAN) plate reader.

**Preparation of SARS-CoV-2-spike pseudoparticles and neutralization assay**. SARS-CoV-2-spike pseudoparticles were obtained by co-transfection of Expi293F™ cells with pCMV-deltaR8.2, pLenti-GFP (Genecopoeia), and pCDNA3.1-SΔC19 according to the manufacturer's instructions (ThermoFisher Scientific) at a ratio of 1:2:1. The supernatant was harvested 72 h post transfection, centrifuged at $1500 \times g$ for 10 min to remove cell debris, and passed through a 0.45 μm filter (LIFEGENE, Israel). Next, the pseudoparticles-containing supernatant was concentrated to 5% of its original volume using Amicon Ultra with 100 KDa cutoff at 16 °C (Merck Millipore). HEK-293 cells stably expressing hACE2 were cultured in Dulbecco's modified Eagle's medium (Gibco) supplemented in 10% fetal bovine serum (FBS), 1% L-glutamine, 1% penicillin streptavidin, and 1% non-essential amino acid. These cells were seeded into 100 μg/ml poly-D-lysine-coated 96-well plates (Greiner) at an initial density of $0.5 \times 10^5$ cells per well. The following day, concentrated pseudo-particles were incubated with two-fold serial dilutions of filtered breastmilk samples starting from dilution of 1:1 (50% breastmilk) or purified IgA

or IgG at a concentration of 50 μg/ml for 30 min at RT and then added to the 96-well pre-seeded plates. After 48 h, the cell medium was replaced with fresh DMEM excluding phenol red. Twenty-four hours later, the 96-well plates were imaged by the IncuCyte ZOOM system (Essen BioScience). Cells were imaged with a 10X objective using the default IncuCyte software settings, which were used to calculate the number of GFP-positive cells from four 488 nm-channel images in each well (data were collected in triplicate). The number of GFP-positive cells was normalized and converted to a neutralization percentage.

**Reporting summary**. Further information on research design is available in the Nature Research Reporting Summary linked to this article.

## Data availability
The processed data generated in this study are provided in the Supplementary Source Data file. The raw data that supports the findings in this study are available under restricted access as they contain information that could compromise research participant privacy/consent, access can be obtained by request from the corresponding author [W.Y]. Source data are provided as a compressed SourceData.zip file. Source data are provided with this paper.

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

## Acknowledgements
This work was supported by the Israel Ministry of Health (MOH) grant #3-17162 [Y.W.] and partially funded by the Israel Science Foundation (ISF) grant 401/18 to [M.D.].

## Author contributions
The study was initiated by M.R.F., A.M., and Y.W. sample collection and processing was carried out by M.R.F., A.K., Y.D. Experiments were carried out by A.K., Y.B., and Y.D. The original manuscript draft was written by M.R.F., A.K., A.M., and Y.W. Neutralization assays using spike-bearing pseudovirus was carried out by M.W., J.A., M.D., and M.G.T. Review and editing of the manuscript were carried out by M.R.F., A.K., Y.B., Y.D., R.L., Y.Y., A.M., and Y.W.

## Competing interests
The authors declare no competing interests.
