## [Peer Review File · Nature Communications]

BNT162b2 mRNA vaccine elicited antibody response in blood and milk of breastfeeding womenReviewers' Comments:

Reviewer #1:

Remarks to the Author:

This manuscript is among the first to assess the IgG and IgA responses against the Spike and RBD of SARS-CoV-2 after vaccination with the Pfizer mRNA vaccine. The paper demonstrates that the vaccine elicits both IgG and IgA responses in milk, though the dominant response in plasma is IgG and the more persistent response in milk is IgG. It also reports on neutralizing capacity of IgG and IgA, but only measures the ACE2 blocking potential of the antibody, which is not measurement of true virus neutralization as measured by the ability to block virus entry into the cell using live virus or a pseudovirus assay. Moreover, the paper claims that it establishes neutralizing IgA response in milk, yet this is measured only with anti-IgG/IgA abs in the ACE2 blocking assay that can have cross reactive potential. And this is not confirmed using IgG depletion studies, which is often required to establish IgA functions in mucosal fluids. Thus, I think additional assays need to be completed to establish the true neutralizing capacity of IgG and IgA elicited by the vaccine. Also, the number of comparisons made between the timepoints of 10 women is quite large and should be reduced and corrected for multiple comparisons.

Reviewer #2:

Remarks to the Author:

In their manuscript, the authors provide evidence of transfer of anti-COVID 19 immunity through breast milk of vaccinated mothers. This information is very important as it might imply that vaccinating breastfeeding mothers broadens the protection to their offspring.

As major comment, I would say that the submitted manuscript appears more as a draft than as a finalised manuscript. I am giving support to this statement here below. While this manuscript was submitted, a work has been published with a similar objective (Gray et al, <https://doi.org/10.1016/j.ajog.2021.03.023>) and therefore limits the novelty of the findings.

Detailed comments:

Format: there is no sub-heading such as intro/ methods/ results/ discussion in the submitted manuscript.

Methodology

Please be more precise in the description of the outcomes of the assays:

" Serum titers measurements and competitive assay" : Could you precise in the title that you are measuring anti Spike and anti RBD IgG and IgA in serum and human milk.

Competitive assay: what do you aim to measure with this assay? Could you describe with more details how you analyse the results

Lactoserum: this word is inappropriate to describe human milk samples unless they have been specifically treated to produce lactoserum. I would use the terminology of aqueous phase of human milk.

Results

Fig.1. Units of antibody levels. I am confused with legends on Y Axis; the Figure legend indicates the values are the endpoint titres. Could the authors explain why Y axis legend is different in panel A versus panel B? In the methods, the authors indicate that serum samples are diluted 100 times while breast milk samples only 2 times. However, titres are quite similar between breast milk samples and serum. Did the authors take into account the dilution factor? This is important in order to be able to compare the magnitude of antibody response in milk versus serum. Please also refer to this figure at the beginning of the description of the results.

Ratio IgG/IgA response. Because the sensitivity of IgA and IgG assays differ, the authors need to

quantify the absolute levels of IgG and IgA antibodies using appropriate standard in order to compare the levels of IgG versus IgA response. I would therefore limit the interpretation of the data and only conclude on the Fig 1 and 2 data that show a decrease in IgA response for 2D14 as compared to 2D7 (is this significant?) while IgG do not decrease. This indicates a higher proportion of IgG is observed along the vaccination process.

Competitive assay. The ref mentioned (11) does not refer to the methods used in the paper. Please explain briefly the principle of the assay and discuss how it differs from virus neutralisation assay. Please avoid using the terminology of neutralising capacity when referring to this competitive assay.

Figure 2. Please delete "antibody neutralising capacity" in the title as this is not shown in that figure. The description of that figure is incomplete in the text.

Discussion

The discussion is extremely limited, and I would recommend to expand to better highlight the importance of the findings and their limitations.

I do not see the relevance of mentioning the possible fecal-oral transmission route except if the authors expect breast milk could only protect the newborn if infected by oral route? Breast milk also covers the oro-pharynx of a newborn and prevent respiratory infections as largely demonstrated for other infectious disease.

Professor Valerie Verhasselt, The University of Western Australia

Response to reviewers:

Reviewer 1:

Reviewer comment: *“This manuscript is among the first to assess the IgG and IgA responses against the Spike and RBD of SARS-CoV-2 after vaccination with the Pfizer mRNA vaccine. The paper demonstrates that the vaccine elicits both IgG and IgA responses in milk, though the dominant response in plasma is IgG and the more persistent response in milk is IgG”.*

Response: We thank the reviewer for the careful reading, accurately understanding the content of the paper in detail, and acknowledging the novelty of the report.

Reviewer comment: *“It also reports on neutralizing capacity of IgG and IgA, but only measures the ACE2 blocking potential of the antibody, which is not measurement of true virus neutralization as measured by the ability to block virus entry into the cell using live virus or a pseudovirus assay. Moreover, the paper claims that it establishes neutralizing IgA response in milk, yet this is measured only with anti-IgG/IgA abs in the ACE2 blocking assay that can have cross reactive potential. And this is not confirmed using IgG depletion studies, which is often to required establish IgA functions in mucosal fluids. Thus, I think additional assays need to be completed to establish the true neutralizing capacity of IgG and IgA elicited by the vaccine.”*

Response: We thank the reviewer for highlighting this important point. We agree that that the ACE2 competition-blocking assay that we used does not entirely reflect the actual measurement of virus neutralization, as measured by the ability to block virus entry. Thus, we performed additional experiments with SARS-CoV-2 spike-bearing pseudovirus and the results have been incorporated into the revised manuscript (main text, and materials and methods) and consolidated in new Figure 3 (the results for the competition-blocking assay have been removed from the manuscript). The neutralization experiments included the isolation of IgG and IgA from breastmilk samples (using protein G and peptide M, respectively) and measuring their neutralization capacity using SARS-CoV-2 spike-bearing pseudovirus. The results of these experiments using the pseudovirus neutralization assay demonstrate that the vaccine-specific breastmilk antibodies do indeed exhibit neutralization capacity.

Fig. 3 Neutralization capacity of breastmilk IgG and IgA. Purified IgG and IgA from breastmilk were subjected to a spike-bearing pseudovirus neutralization assay; percent inhibition was calculated by measuring GFP-positive HEK-293 cells stably expressing hACE2. The number of

GFP-positive cells was normalized and converted to a neutralization percentage. Each sample was measured in triplicate, and bar graph plots present mean values \pm SEM.

Reviewer comment: “Also, the number of comparisons made between the timepoints of 10 women is quite large and should be reduced and corrected for multiple comparisons.”

Response: We thank to reviewer for this important comment. We stress that the analysis results that are summarized in Figure 1 (and supplementary Figure 3 for anti-RBD) aimed to provide data regarding the antibody levels at each time point compared to the control group (pre-pandemic independent samples). As such, we used statistical analysis to determine the significance of the titers at each time point, aiming to identify the earliest time point that the titers were significantly higher than those in the control group and to further evaluate whether the significant difference is sustained in the subsequent time points. As we used multiple comparisons, the P values used were subjected to the stringent Bonferroni correction (accounting for multiple comparisons). For the anti-spike titers (Figure 1) the 1D14 time point was found to be the first time point that showed endpoint titers for IgG and IgA that were significantly higher than those in the control group and the additional time points retained this significant difference; thus, no further comparisons beyond 1D14 are presented in Figure 1 for IgG and IgA in both serum and breastmilk.

A similar revision was made in Supplementary Figure 3 showing the anti-RBD levels. Here, we observed that the first time point with a marked increase of IgA was 2D7.

We revised the text in the manuscript to describe these changes: “On 1D7, the spike-specific endpoint titers of IgG and IgA in the serum and breastmilk had not increased significantly above the titers in the control group (pre-pandemic serum and breastmilk), and the first significant increase in antibody titers was evident on 1D14. The titers for the vaccinated group remained significantly higher than those of the control group, with spike-specific endpoint titers peaking on 2D7, followed by a non-significant decrease on 2D14. A similar trend was observed for the RBD-specific IgG titers (Supplementary Fig. 3). The RBD-specific IgA titers, in both serum and breastmilk, reached significant levels vs. control titers on 2D7, suggesting that the accumulation of IgA antibodies against the restricted region of the spike (i.e., RBD) was slightly delayed compared to IgG, both in serum and breastmilk.”

To emphasize that the endpoint titers for anti-spike/RBD were similar to the anti-tetanus toxoid (TT) titers, we kept the comparison of the last time point of breastmilk antibodies to the titers of anti-TT, thus showing that there is no significant difference between the antibody titers (based on p values after Bonferroni correction).

We further examined the reviewer’s comment in the context shown in Figure 2. The analysis in Figure 2 aimed to reveal the dynamics of the vaccine-specific titers in comparison to the first time point used as the reference time point. As the objective here was the evaluate the dynamics at high resolution, we performed the analysis based on the comparison of all time points. This is important, as this is the first time the dynamics of breastmilk antibodies is reported for the mRNA-vaccine; thus, we did not have an a-priori hypothesis for the behavior of the dynamics. Here, too, multiple comparison analysis was corrected by applying the Bonferroni correction.

Reviewer 2:

Reviewer comment: *"In their manuscript, the authors provide evidence of transfer of anti-COVID 19 immunity through breast milk of vaccinated mothers. This information is very important as it might imply that vaccinating breastfeeding mothers broadens the protection to their offspring."*

Response: We thank the reviewer for appreciating the importance of our study and recognizing the potential impact of our findings.

Reviewer comment: *"As major comment, I would say that the submitted manuscript appears more as a draft than as a finalised manuscript. I am giving support to this statement here below."*

Response: We believe that by addressing all comments raised by the reviewers the manuscript has been improved substantially. Of note, we aimed to keep the report concise and to focus to the extent possible on highlighting new findings regarding the sub-population of lactating women that was not included in the vaccine clinical trials.

Reviewer comment: *"While this manuscript was submitted, a work has been published with a similar objective (Gray et al, <https://doi.org/10.1016/j.ajog.2021.03.023>) and therefore limits the novelty of the findings."*

Response: We thank the reviewer for pointing out this important work. That report had not been published at the time of our submission, but we have now cited this paper in the revised manuscript and have also included an additional paper that was published since our manuscript was submitted (Perl, S. H. et al., <https://doi:10.1001/jama.2021.5782>). The above notwithstanding, we believe that the current work still has a high degree of novelty and impact for the following reasons: First, we provide data regarding the synchronization between the antibody response in the blood and breastmilk, which was not reported in the above-mentioned reports. Second, we provide evidence regarding the neutralization capacity of the breastmilk antibodies, which was also not included the recent reports. Third, the vaccine-specific antibodies titers were compared to tetanus toxoid (TT) specific antibodies elicited in the same cohort following Tdap maternal vaccination. This comparison is of high importance as we show that the COVID-19 vaccine elicits breastmilk antibodies that do not differ significantly in comparison to the antibodies elicited following the administration of the well-established Tdap vaccine.

Reviewer comment: *"Format: there is no sub-heading such as intro/ methods/ results/ discussion in the submitted manuscript."*

Response: We revised the manuscript and now it includes sub-headings in accordance to the journal's guidelines.

Reviewer comment: *"Methodology: Please be more precise in the description of the outcomes"*

of the assays: ‘Serum titers measurements and competitive assay’ : Could you precise in the title that you are measuring anti Spike and anti RBD IgG and IgA in serum and human milk.”

Response: We revised the title to be more precise in the description of the assay outcomes. The title after revision is: “Endpoint titers of spike and RBD-specific IgA, IgG in serum and breastmilk”.

Reviewer comment: “Competitive assay: what do you aim to measure with this assay? Could you describe with more details how you analyse the results.”

Response: We thank the reviewer for highlighting this important point. We agree that that the ACE2 competition-blocking assay we used does not entirely reflect the actual measurement of the virus neutralization as measured by the ability to block virus entry. Thus, we performed additional experiments with SARS-CoV-2 spike-bearing pseudovirus and incorporated the results into the revised manuscript (main text, material and methods and new Figure 3) (we deleted the results based on the competition-blocking assay). The neutralization experiments included the isolation of IgG and IgA from breastmilk samples (using protein G and peptide M, respectively) and measuring their neutralization capacity using SARS-CoV-2 spike-bearing pseudovirus. The results of these experiments demonstrate that the vaccine-specific breastmilk antibodies exhibit neutralization capacity.

Fig. 3 Neutralization capacity of breastmilk IgG and IgA. Purified IgG and IgA from breastmilk were subjected to a spike-bearing pseudovirus neutralization assay, and percent inhibition was calculated by measuring GFP-positive HEK-293 cells stably expressing hACE2. The number of GFP-positive cells was normalized and converted to a neutralization percentage. Each sample was measured in triplicate, and bar graph plots present the mean values \pm SEM.

Reviewer comment: “Lactoserum: this word is inappropriate to describe human milk samples unless they have been specifically treated to produce lactoserum. I would use the terminology of aqueous phase of human milk.”

Response: We thank the reviewer for highlighting the inappropriate usage of the term lactoserum. We have revised the manuscript and replaced the word lactoserum with breastmilk.

Reviewer comment: *“Fig.1. Units of antibody levels. I am confused with legends on Y Axis; the Figure legend indicates the values are the endpoint titres. Could the authors explain why Y axis legend is different in panel A versus panel B? ”*

Response: We thank the reviewer for pointing out the unclear designations of the axis in Figure 1 and apologize for creating confusion.

The Y-axis in panel A of Figure 1 is given is on the logarithmic scale and in panel B on the linear scale. The figure legend states this difference. We used the linear scale in panel B as it visually enabled the emphasis of the endpoint titer for each donor separately, thus, highlighting the variability between the donors.

Reviewer comment: *“In the methods, the authors indicate that serum samples are diluted 100 times while breast milk samples only 2 times. However, titres are quite similar between breast milk samples and serum. Did the authors take into account the dilution factor? This is important in order to be able to compare the magnitude of antibody response in milk versus serum. Please also refer to this figure at the beginning of the description of the results.”*

Response: We thank the review for the careful reading and accurately understanding the details of the analysis. The different dilution factors of the samples from serum and breastmilk were taken into account. We would like to highlight that the Y-Axis range in the graphs of serum are different from those for the breastmilk. This applies to the Y-Axis units in Figure 1 and Figure 2 as well as Supplementary Fig. 2, Supplementary Fig. 3 and Supplementary Fig. 5. We agree that this way of presentation may be misleading, but alignment of the Y-Axis range for breastmilk titers to the Y-Axis range of serum resulted in a “packed” graph that precluded easy visualization of the plotted results. Of note, the Y-axis range for both isotypes (i.e., IgG and IgA) within each compartment (i.e., serum and breastmilk) were the same throughout all above-mentioned figures.

As the reviewer suggested we revised the manuscript to include the reference to the figure at the beginning of the description of the results.

Reviewer comment: *“Ratio IgG/IgA response. Because the sensitivity of IgA and IgG assays differ, the authors need to quantify the absolute levels of IgG and IgA antibodies using appropriate standard in order to compare the levels of IgG versus IgA response. I would therefore limit the interpretation of the data and only conclude on the Fig 1 and 2 data that show a decrease in IgA response for 2D14 as compared to 2D7 (is this significant?) while IgG do not decrease. This indicates a higher proportion of IgG is observed along the vaccination process.”*

Response: We totally agree that the interpretation of the data regarding the IgG/IgA should be limited due to the different sensitivities of the IgG and IgA assays. To overcome this limitation, we purified IgG and IgA to be used as a standard curve, thereby eliminating the bias that may be introduced due the different sensitivities of the assays. Supplementary Fig. 4 was replaced with a new figure, and the Materials and Methods section was revised to include the description of the experimental setup. We believe that the new assay using a standard curve supports the statement regarding the vaccine-specific IgG:IgA antibodies and the ratio is now shown as a “Molar ratio”.

Competitive assay. The ref mentioned (11) does not refer to the methods used in the paper. Please explain briefly the principle of the assay and discuss how it differs from virus neutralisation assay. Please avoid using the terminology of neutralising capacity when referring to this competitive assay.

Response: We thank the reviewer for highlighting this important point and have addressed this issue as described above in the response to reviewer comment: *“Competitive assay: what do you aim to measure with this assay? Could you describe with more details how you analyse the results.”*. As we do not present data derived from the competition-blocking assay we removed ref 11 from the manuscript.

Reviewer comment: *“Figure 2. Please delete ‘antibody neutralising capacity’ in the title as this is not shown in that figure. The description of that figure is incomplete in the text.”*

Response: We thank the reviewer for pointing out this omission. We have revised the legend to Figure 2 and deleted the words: “antibody neutralization capacity”. Additionally, we have revised the manuscript accordingly.

Reviewer comment: *“Discussion: The discussion is extremely limited, and I would recommend to expand to better highlight the importance of the findings and their limitations.”*

Response: We thank the reviewer highlighting this point. We have expanded the discussion to stress the importance of the findings and their limitations.

Reviewer comment: *“I do not see the relevance of mentioning the possible fecal-oral transmission route except if the authors expect breast milk could only protect the newborn if infected by oral route? Breast milk also covers the oro-pharynx of a newborn and prevent respiratory infections as largely demonstrated for other infectious disease.”*

Response: We agree with the reviewer that the relevance of mentioning the possible fecal-oral transmission route is not the only transmission route. We have revised the manuscript accordingly.

Reviewers' Comments:

Reviewer #1:

Remarks to the Author:

The revised manuscript addresses a lot of reviewer concerns and improved the presentation and the analysis of the binding antibody responses. However, a major critique from both reviewers was the lack of neutralization data. While pseudovirus neutralization data from purified IgG and IgA was added, it seems inadequate for publication. It appears to have been run at a single point dilution, while the standard for neutralization assessments to run the assays across several dilutions. Moreover, neutralization is typically only reported if the responses are above 50%. In the graph presented, many responses are under 50% or around 50%. Also, when you have purified ab, the result would be better presented either as an inhibitory concentration 50% or at least normalize the response for the magnitude of the Ig content. Moreover, milk samples can also have cellular toxicity and so some assessment of cell toxicity would be important. Thus, while the edits improved the binding ab results, the neutralization assessment is still inadequate.

Reviewer #2:

Remarks to the Author:

I thank the authors for the additional experiments they performed and revision of the manuscript that has much improved in terms of clarity and novelty. I have some comments to add:

Neutralising assay: I thank the authors for adding those data that are very interesting and totally novel. It would be nice that the authors indicate what amount of antibody was added to the pseudo-virus neutralising assay in the result and method section instead of discussion. I would also indicate in the result section that antibodies were purified from 2D7 milk samples. I would like to share some thoughts regarding the interpretation of the data:

-Even though IgA neutralising activity is about twice lower than IgG in the assay, breast milk contains about 10 times more Spike specific IgA than IgG (supplemental Fig.4). Therefore, neutralising activity of breast milk may rely more on IgA than IgG. This could be discussed. Also, if the authors have the data, it could be informative to compare the neutralising activity of IgA versus IgG antibody-depleted milk (and both IgG and IgA depleted milk)

-From the data been obtained with total antibody purified from 2D7 milk samples, the authors concluded that vaccination induced neutralising antibody. However, I am not sure we can conclude this as there is no ctrl in the assay with IgG and IgA purified from ctrl milk, that may already exert a non-specific inhibitory activity. I think it is important to add that ctrl (or data with Ig purified from 1D7 milk). In the same line, I think the authors can't indicate in the discussion and abstract that breastmilk vaccine-specific IgG and IgA have neutralising activity, as the assay tested the neutralising activity of total IgG and IgA.

IgG/IgA molar ratio: "The IgG:IgA ratio in breastmilk indicated that the vaccine-specific response was not dominated by IgA, but rather that the IgG:IgA molar ratio increased over time, as previously described following respiratory syncytial virus immunization¹²."

When looking at Fig 1 and Supplemental Fig 4, I am a bit confused:

- Serum immune response looks like largely dominated by IgG in Fig.1, however the ratio IgG/ IgA is just above 1 at 2D7 time point. Could you clarify this?

- Milk response is largely dominated by IgA, as shown by ratio in Sup Fig 4, that is > 10 times for IgA versus IgG. Thus, I would not conclude that response in breast milk is dominated by IgG, I would say that IgG response increases over time.

I thank the author for their work that will be very informative for vaccine recommendation for breastfeeding mothers.

With my best regards,
Prof Valerie Verhasselt

Reviewer #1 (Remarks to the Author):

Reviewer comment: *“The revised manuscript addresses a lot of reviewer concerns and improved the presentation and the analysis of the binding antibody responses.”*

Response: We thank the reviewer for acknowledging and appreciating that the changes we have included in the revised manuscript addresses a lot of the concerns. We thank the reviewer for her/his valuable suggestions that helped to improve the presentation and the analysis of the binding antibody responses.

Reviewer comment: *“However, a major critique from both reviewers was the lack of neutralization data. While pseudovirus neutralization data from purified IgG and IgA was*

added, it seems inadequate for publication. It appears to have been run at a single point dilution, while the standard for neutralization assessments to run the assays across several dilutions.”

Response: We appreciate the constructive criticism and agree with the reviewer that it would be better to assess the neutralization based on assays across several dilutions. We made a genuine effort to address this concern. However, the main challenge we encountered was the restricted amount of breastmilk that was generously donated by the study participants. The small amount of collected breastmilk was not sufficient for the purification of IgG/IgA in adequate amount and/or concentration to carry out the assays across dilutions. Nevertheless, we were able to design and conduct a set of pseudovirus neutralization experiments across breastmilk dilution series that enabled the calculation of breastmilk IC₅₀ and these data are included in Fig. 3a and Fig. 3b of the revised manuscript.

Reviewer comment: *“Moreover, neutralization is typically only reported if the responses are above 50%. In the graph presented, many responses are under 50% or around 50%. Also, when you have purified ab, the result would be better presented either as an inhibitory concentration 50% or at least normalize the response for the magnitude of the Ig content.”*

Response: As mentioned in the previous response, the restricted amount of breastmilk precluded our ability to provide neutralization assessment based on serial dilution. Still, we were able to generate data that supports the calculation of the IC₅₀ for IgG from four breastmilk samples (see Figure attached – for reviewers only). As we were not able to generate such data for all samples and for both IgG and IgA, we decided not to include the data in the manuscript. Thus, we have added a section in the manuscript named “Limitation of study”) where we highlight this limitation. Additionally, we have considered to remove the Figure of purified IgG/IgA neutralization at a single concentration. However, we believe that the data is important to demonstrate the variability of the neutralizing antibody response across patients. As suggested by the reviewer we have normalized the % inhibition for the magnitude of the Ig content (% inhibition per 5ug antibody) and this data is included in Fig. 3c-d of the revised manuscript.

Breastmilk IgG neutralization assays (Reviewers only)

Reviewer comment: *“Moreover, milk samples can also have cellular toxicity and so some assessment of cell toxicity would be important.”*

Response: We thank the reviewer for pointing out the importance of cell toxicity information. We now provide data on the cell toxicity using XTT assay (new Supplementary Fig. 7).

Reviewer comment: *“Thus, while the edits improved the binding ab results, the neutralization assessment is still inadequate.”*

Response: We thank the reviewer for the constructive comments and believe that based on the data included in the revised manuscript it is within reason to state that breastmilk following mRNA vaccine exhibits antibody neutralization activity.

Reviewer #2 (Remarks to the Author):

Reviewer comment: *“I thank the authors for the additional experiments they performed and revision of the manuscript that has much improved in terms of clarity and novelty.”*

Response: We would like to thank the reviewer for appreciating the additional experiments we performed and revisions we have incorporated in the manuscript. We also would like to

recognize the contribution of the reviewer for her suggestion that facilitated the improvement of the report in terms of clarity and novelty.

Reviewer comment: *"I have some comments to add: Neutralising assay: I thank the authors for adding those data that are very interesting and totally novel. It would be nice that the authors indicate what amount of antibody was added to the pseudo-virus neutralising assay in the result and method section instead of discussion. I would also indicate in the result section that antibodies were purified from 2D7 milk samples."*

Response: We thank the reviewer for these important comments. We have added to the result section the amount of antibody that was used in the pseudo-virus neutralization assay and indicated that these antibodies were purified from 2D7 milk samples.

Reviewer comment: "I would like to share some thoughts regarding the interpretation of the data:

-Even though IgA neutralising activity is about twice lower than IgG in the assay, breast milk contains about 10 times more Spike specific IgA than IgG (supplemental Fig.4). Therefore, neutralising activity of breast milk may rely more on IgA than IgG. This could be discussed."

Response: This is a very interesting point. While IgA is the dominant isotype in breastmilk, the presence of IgG is not completely clear as it cannot be transported to the breastmilk via the polymeric Ig receptor (pIgR) by transcytosis. We hypothesize that IgG is generated *in situ* by resident plasma B cells and that the generated IgG has a role in the protection from local infections. We agree that breastmilk IgA levels are higher than breastmilk IgG and added a sentence in the discussion section of the revised manuscript addressing this point, along with the increased reliance on IgA in term of neutralization activity.

Reviewer comment: "Also, if the authors have the data, it could be informative to compare the neutralising activity of IgA versus IgG antibody-depleted milk (and both IgG and IgA depleted milk)."

Response: The isolation IgG and IgA from serum and breastmilk was carried out by sequential affinity chromatography steps thus, first IgG was isolated using protein G followed by the isolation of IgA from the flow through of the protein G column. Thus, we did not obtain independent breastmilk samples that were either IgG or IgA depleted. Moreover, the amount of breastmilk was very limited and hence, we were not able to generate new data based on breastmilk samples.

Reviewer comment: "-From the data been obtained with total antibody purified from 2D7 milk samples, the authors concluded that vaccination induced neutralising antibody. However, I am not sure we can conclude this as there is no ctrl in the assay with IgG and IgA purified from ctrl milk, that may already exert a non-specific inhibitory activity. I think it is important to add that ctrl (or data with Ig purified from 1D7 milk). In the same line, I think the authors can't indicate in the discussion and abstract that breastmilk vaccine-specific IgG and IgA have neutralising activity, as the assay tested the neutralising activity of total IgG and IgA."

Response: We agree with the reviewer that controls are very important to conclude that the vaccination induced the generation of neutralizing antibodies as IgG and IgA isolated from control samples may exert non-specific activity. It seems that we did not emphasize the presence of controls in the figure as the experiment we conducted included control samples that were obtained from individuals during 2018 (prior to the pandemic). The data is shown in Figure 3c. We have also revised the manuscript to include a statement that will emphasize the inclusion of control samples in the assay. Moreover, we revised the manuscript so it will not include a statement that vaccine-specific IgG and IgA have neutralization activity, rather, we emphasize the neutralization activity of breastmilk.

Reviewer comment: “IgG/IgA molar ratio: “The IgG:IgA ratio in breastmilk indicated that the vaccine-specific response was not dominated by IgA, but rather that the IgG:IgA molar ratio increased over time, as previously described following respiratory syncytial virus immunization¹².” When looking at Fig 1 and Supplemental Fig 4, I am a bit confused: - Serum immune response looks like largely dominated by IgG in Fig.1, however the ratio IgG/ IgA is just above 1 at 2D7 time point. Could you clarify this?”

Response: We thank the reviewer for highlighting this important point that needs to be clarified. Indeed, Supplemental Fig. 4 is misleading and actually after carefully examining the original graph (prism GraphPad version 9), we noticed that the numbering for the Y-axis is not aligned with the actual data in the graph. This is probably due to the Prism’s feature that enables to use custom axis numbering. We have re-plotted the figure and disabled the feature and revised graph is now included in the revised version of the supplementary document (new Supplemental Fig. 4). In the new Supplemental Fig. 4 for the readers convenience, we included ratio plots by compartment (i.e. serum and breastmilk). We also would like to note that the Y – axis is in the log scale. We sincerely apologize for this and thank the reviewer going into the details and enabling us to improve the representation of the data.

Reviewer comment: “- Milk response is largely dominated by IgA, as shown by ratio in Sup Fig 4, that is > 10 times for IgA versus IgG. Thus, I would not conclude that response in breast milk is dominated by IgG, I would say that IgG response increases over time.”

Response: We thank the reviewer for highlighting this point. We agree that it is more suitable to describe the temporal dynamics of IgG/IgA ratio by stating that the IgG response increases over time. We have revised the manuscript to reflect this important insight.

Reviewer comment: “I thank the author for their work that will be very informative for vaccine recommendation for breastfeeding mothers.”

Response: We thank the reviewer for her positive feedback and for acknowledging that the data we present will be very informative for vaccine recommendation for breastfeeding mothers. We thank the reviewer for her insightful comments and suggestions that helped to improve substantially the manuscript.

Reviewers' Comments:

Reviewer #1:

Remarks to the Author:

The added neutralization data is useful and gets closer to determining the true neutralization capacity of breast milk after mRNA vaccination. However, neutralization by IgA is not confirmed by this work, so I would caution against that conclusion in the manuscript. In Fig 3b, the unit is referred to as IC50 when it should be inhibitory dilution 50%, therefore ID50.

Also, I would not present % less than 50% at a single dilution as true neutralization, so I would draw a dotted line on 50% or remove the data.

Also, units need to be added or clarified to all figures (such as Fig 1, there is no unit on the graphs, and Fig 3 % needs to be added to the inhibition graph)

Reviewer #1 (Remarks to the Author):

Reviewer comment: “The added neutralization data is useful and gets closer to determining the true neutralization capacity of breast milk after mRNA vaccination.”

Response: We thank the reviewer for acknowledging and appreciating that the added neutralization data is useful and gets closer to determining the true neutralization capacity of breast milk after mRNA vaccination.

Reviewer comment: “However, neutralization by IgA is not confirmed by this work, so I would caution against that conclusion in the manuscript.”

Response: We understand the reviewers’ concern regarding the neutralization by IgA. To address this concern, we added to the “*limitation of the study*” section the following sentence: “Specifically, as IgA at the concentration of 50 µg/ml did not exceed the 50% inhibition, we are careful with any statement regarding the neutralization capacity of IgA.”

Reviewer comment: “In Fig 3b, the unit is referred to as IC50 when it should be inhibitory dilution 50%, therefore ID50.”

Response: We agree that the appropriate units should be ID₅₀ thus, revised Fig.3a- Fig. 3b and any reference in the text that includes IC50.

Reviewer comment: “Also, I would not present % less than 50% at a single dilution as true neutralization, so I would draw a dotted line on 50% or remove the data.”

Response: We agree with the reviewer and per the reviewer suggestion we revised the figure to include a dashed line at 50% inhibition.

Reviewer comment: “Also, units need to be added or clarified to all figures (such as Fig 1, there is no unit on the graphs, and Fig 3 % needs to be added to the inhibition graph)”

Response: We have revised the figures to include or clarify the units.